# Projecting and Forecasting the Latent Volatility for the Nasdaq OMX Nordic/Baltic Financial Electricity Market Applying Stochastic Volatility Market Characteristics

**Per Bjarte Solibakke** 

Faculty of Economics and Management, Norwegian University of Science and Technology, Larsgårdsveien 2, 6025 Ålesund, Norway; per.b.solibakke@ntnu.no; Tel.: +47-7016-1427 or +47-9003-5606

**Abstract:** In this empirical study, multifactor stochastic volatility models for the financial Nordic/Baltic power markets are developed, implemented, and analyzed. Stochastic volatility projections are the primary aim, followed by volatility forecasts and market repercussions. The research provides a functional variant of the conditional distribution ($f(x|y)$) based on conditional moments and a long-simulated state vector realization (MCMC-GMM) that is evaluated on observed data (a nonlinear Kalman Filter) and applicable for step-forward volatility forecasts. For front year and quarter financial electricity contracts, the SV model creates two mean-reverting factors: one persistent and slowly moving component and one choppy, rapidly moving component. According to these factors, static volatility predictions with optimum and generous lags have a Theil covariance percentage of well over 97 percent for the front year contracts and 86 percent for the front quarter contracts. The volatility visibility and its associated static forecasts improve market transparency and will eventually make diversification and risk management easier to implement.

**Keywords:** forecasting; Markov Chain Monte Carlo (MCMC-GMM) estimation; nonlinear Kalman filter; stochastic volatility

## 1. Introduction

This research creates and tests multifactor scientific stochastic volatility (SV) models for predicting future electricity market volatility, which is characterized by extremes and unpredictability. Electricity volatility is a measure of the spread around the mean return on financial energy contracts. Given that electricity markets are influenced by a variety of factors such as weather, local economic activity, the global financial outlook, international prices, resource availability, investment in future resources, government policies, and the physical and mechanical constraints on plant or infrastructure, periods of high volatility are frequently the rule rather than the exception, making budgeting and cost control difficult. Understanding and forecasting the effect of major market fundamental risk variables is therefore critical for all market players since volatility is expected to persist. Volatility is low (high) when daily price fluctuations are firmly bunched together (spread apart). Volatility measurements provide predictive properties for future returns, and volatility models have therefore been used globally to anticipate the absolute size of returns, quantiles, and full densities. Squared price changes, for example, are a basic and often used indicator for financial market volatility.

One of the characteristics that distinguishes market volatility is that it is not visible (latent) for market participants. The unobservability makes it difficult to assess the predictive efficacy. Models are therefore difficult to evaluate. Simultaneously, volatility prediction and understanding the empirical features of future contract pricing are critical for developing risk management techniques for portfolio selection, derivatives and hedging, market making, and market timing. The Nasdaq OMX financial market also offers a liquid market for derivatives on future contracts. Consequently, any Nasdaq OMX market participant

will benefit from a volatility model that predicts volatility. For instance, a successful risk manager must be able to forecast if his portfolio will increase or decline in the future. For hedging purposes, a risk manager will benefit from knowing the volatility as a contract approaches maturity. Volatility, for example, is the sole parameter that requires estimate in the Black–Scholes formula. In an energy market, an option trader will want to know how volatile the contract will be over the contract's lifetime. Derivatives encourage hedging, a risk-reduction approach that necessitates a thorough understanding of how to value derivatives and which risks should and should not be hedged. To hedge a contract, a trader will need to know the expected volatility. The volatility estimates may also be useful in calculating binomial model parameters (*u* and *d*). In general, higher (lower) volatility raises (decreases) derivative prices. As a result, if market players anticipate a drop (increase) in volatility, they will sell (purchase) call and put option contract holdings that are not part of speculative or hedging positions. A portfolio manager may desire to sell an asset or an asset portfolio before it becomes very volatile. According to worldwide portfolio and asset research, when volatility grows, so does risk, and portfolio and asset movements fall. If a portfolio manager adds extra assets to his portfolio, the additional assets diversify the portfolio if they do not covary (correlation less than 1) with the other assets in the same portfolio. As a result, portfolios frequently recommend diversification, highlighting the need of asset allocation. Furthermore, if a market maker expects that future volatility will vary, he may alter his bid-ask market spread, knowing that when volatility rises (falling), the bid-ask spread normally rises (falls).

The main stylized features of asset, currency, and commodity price variations may be described using stochastic volatility models, which have a basic and uncomplicated structure [1]. The observed frequently and regularly fluctuating volatility motivates stochastic volatility. Market participants who understand volatility's dynamic behavior are more likely to have realistic expectations about future prices and the risks to which they are exposed [2]. In financial markets, time-varying volatility is widespread, and market players who grasp the dynamic nature of volatility are more likely to have correct expectations about future prices. The SV implementation tries to depict the evolution of volatility over time. Volatility, although being a non-traded instrument with inaccurate estimates, may be thought of as a latent variable that can be modeled and predicted by its direct influence on the magnitude of returns. Because risks can alter in various ways over time, multi-factor stochastic models for volatility temporal development are required. Thus, the use of SV models is essentially motivated by three criteria. First, the number of events on day *t* is unpredictable [2]. The number of day *t* events is proportional to the SV methodology. Second, the trading clock (time deformation) runs at varying intensities on different days, and the clock is frequently represented by trade volume [3]. Finally, Hull and White [4] show that SV models for continuous volatility variables provide a decent approximation to diffusion processes (closely related to realized variance). In contrast, general autoregressive conditional heteroscedasticity (GARCH) processes, which are commonly referred to as SV, do not use this terminology. These models explicitly model the conditional variance given the econometrician's prior observed returns.

The approach adapts Chernozhukov and Hong's MCMC-GMM estimator [5] for stochastic situations, which is reported to be much superior than typical derivative-based hill climbing optimizers. The main reason is asymptotically correct standard errors for the weighting matrix $\left(\widetilde{I}_n\right)$. Furthermore, when the structural models are accurately stated, the normalized value of the objective function is asymptotically $\chi^2$ distributable (and the degrees of freedom are specified). Gallant and McCulloch [6] and Gallant and Tauchen [7,8] implemented multivariate statistical models built from scientific concerns using the Bayesian Markov Chain Monte Carlo (MCMC) modeling approach. The methodology is a systematic approach to obtaining moment conditions for a structural model's parameters using the generalized method of moments (GMM) estimator [8]. The Chernozhukov and Hong [9] estimator was used to conserve model parameters in the range where projected shares are positive for each observed price/expenditure vector. Further-

more, the technique takes into account limitations, inequality constraints, and useful prior information (on the model parameters and functions).

The main results show that the MCMC-GMM estimated multifactor SV model, extended with a non-linear Kalman filter, visualize and reproject the latent volatility. Knowing that volatility is strongly negatively correlated with commodity prices, market strategies involving volatility will enhance diversification as well as insure market participants against market crashes. Furthermore, static volatility forecasts can make market strategies even more accurate. The rest of the article is organized as follows. Section 2 describes the methodology and explicitly describes the non-linear Kalman filter. Section 3 characterizes the Nasdaq OMX front year and front quarter contracts. Section 4 reports the empirical results. Section 5 discusses findings for the electricity market, and Section 5 summarizes and concludes the paper.

## 2. Literature and Methodologies

Rather than specifying the predictive distribution of price returns directly, the SV technique does so indirectly, using the model's structure. Because the SV model has its own stochastic process, the econometrician is not concerned with the anticipated one-step-ahead distribution of returns collected over an arbitrary time interval. The application of Andersen et al. [10] is used as a starting point, with the known stochastic volatility diffusion for an observed stock price $S_t$ provided by

$$\frac{dS_t}{S_t} = (\mu + c(V_{1,t} + V_{2,t}))dt + \sqrt{V_{1,t}}dW_{1,t} + \sqrt{V_{2,t}}dW_{2,t}$$

where the unobserved volatility processes $V_{i,t}$, $i = 1, 2$, are either log linear or square root (affine). The $W_{1,t}$ and $W_{2,t}$ are standard Brownian motions that are possibly correlated with $(dW_{1,t}, dW_{2,t})$. Andersen et al. [10,11] evaluated both variants of the stochastic volatility model using daily S&P500 stock index data from 1953 to 31 December 1996. Both variants of the SV model were categorically rejected. Adding a jump component to a simple SV model, on the other hand, significantly improves the fit, owing to two well-known characteristics: fat non-Gaussian tails and persistent time-varying volatility. Chernov et al. [12] used an SV model with two stochastic volatility variables and found promising results. The authors look at two different types of settings for the volatility index functions and factor dynamics: affine and logarithmic. The models are based on daily Dow Jones Index data from 2 January 1953, through 16 July 1999. They discover that models with two volatility variables perform significantly better than models with simply one. One of the volatility variables is very stable, whereas the other is choppy and mean reverting. Solibakke [13] implements a multifactor logarithmic stochastic volatility model for the European equity markets. Applying continuous time price movements ($y_t$), the logarithmic model is applicable for commodities, (crypto-)currencies, equities, and interest rates/bonds. The unique model is specified below using two stochastic volatility factors. (See Solibakke [14] for a detailed definition and specification of a two-factor stochastic volatility model. See also [15]).

$$
\begin{aligned}
y_t &= a_0 + a_1(y_{t-1} - a_0) + \exp(V_{1t} + V_{2t}) \cdot u_{1t} \\
V_{1t} &= b_0 + b_1(V_{1,t-1} - b_0) + u_{2t} \\
V_{2t} &= c_0 + c_1(V_{2,t-1} - c_0) + u_{3t} \\
u_{1t} &= dW_{1t} \\
u_{2t} &= s_1\left(r_1 \cdot dW_{1t} + \sqrt{1 - r_1^2} \cdot dW_{2t}\right) \\
u_{3t} &= s_2\left(\frac{r_2 \cdot dW_{1t} + \left((r_3 - (r_2 \cdot r_1))/\sqrt{1 - r_1^2}\right) \cdot dW_{2t} +}{\sqrt{1 - r_2^2 - \left((r_3 - (r_2 \cdot r_1))/\sqrt{1 - r_1^2}\right)^2} \cdot dW_{3t}}\right)
\end{aligned}
\tag{1}
$$

$W_{i,t}$, $i = 1, 2$, and 3 are standard Brownian motions (random variables). The parameter vector is $\theta = (a_0, a_1, b_0, b_1, c_0, c_1, s_1, s_2, r_1, r_2, r_3)$. The $r$'s are Cholesky decomposition correlation coefficients that enforce an internally consistent variance/covariance matrix. In this study, the logarithmic model with numerous stochastic volatility variables is adopted [12,14]. In the model, the Cholesky decomposition for consistency is employed to enhance correlation between the model elements. The main justification for using correlation modeling is that it allows for the introduction of asymmetry effects (correlation between return and volatility innovations). Rosenberg [16], Clark [3], Taylor [2], and Tauchen and Pitts [17] are early mentions. Gallant et al. [18], Andersen [10,19], Durham [20], Shephard [15], and Chernov et al. [12] are more recent references.

For statistical analysis of a stochastic volatility model produced from a scientific process, the article employs a computational technique described by Gallant and McCulloch [6] and Gallant and Tauchen [7,8]. The technique may be stated intuitively as follows. To begin, a reduced-form auxiliary model (ARMA-GARCH) with a general parameterization is estimated to provide a tractable likelihood function. The phase gathers significant information about the probabilistic structure of the data sample from the conditional model and estimated set of score moment functions $(f(y|x))$. Second, the logarithmic stochastic volatility model is estimated using realistic starting values for the model coefficients. The optimal MCMC-GMM methodology uses a long-simulated sample (>100 k) for the continuous time SV model described above. Parameters are changed using the Metropolis–Hastings method and parallel computation (OpenMPI, https://www.open-mpi.org (accessed on 1 April 2022)) to provide the best fit to the quasi-score moment functions assessed on the simulated data. A comprehensive collection of model diagnostics and a clear metric for gauging the level of SV model success are helpful byproducts. As already shown, the scientific stochastic volatility model can be easily simulated, but it cannot generate likelihoods. Finally, reprojection [21], a computationally expensive, simulation-based non-linear Kalman filter approach, works backward from the observed process to infer the unobserved state vector. (A Kalman filter [22] is an algorithm for sequentially updating a projection for a dynamic system. The algorithm provides a way to calculate exact finite-sample forecasts). That is, from conditional moments and the optimally estimated SV model $(\phi = \hat{\phi})$, a by-product is a long-simulated realization of the volatility state vector $\{\hat{V}_{i,t}\}_{t=1}^N$, $i = 1, 2$ and the corresponding returns $\{\hat{y}_t\}_{t=1}^N$. The simulation $\{\hat{y}_t\}_{t=1}^N$ and $\{\hat{V}_{i,t}\}_{t=1}^N$, $i = 1, 2$ makes it possible to calibrate the functional form of the conditional distribution of these volatility functions. For this calibration, an SNP model is re-estimated on the simulated returns $\hat{y}_t$, and remembering that the model provides a convenient representation of the step-ahead conditional variance $\hat{\sigma}_t^2$ of simulated returns $\hat{y}_{t+1}$ given the long simulated returns. Ordinary regressions are run of $\hat{V}_{i,t}$, $i = 1, 2$ on $\hat{\sigma}_t^2$, $\hat{y}_t |\hat{y}_t|$ with generously long lags of these series. The functions are then simply evaluated on the observed data series $\{\tilde{y}_t\}_{t=1}^n$. Generally, from a large data set, the conditional distributions of functions of $V_{i,t}$, $i = 1, 2$ given $\{\hat{y}_\tau\}_{\tau=1}^t$ through non-linear Kalman filtering [22], the functions are evaluated on the observed data $\{\tilde{y}_\tau\}_{\tau=1}^t$, giving volatility values $\tilde{V}_{i,t}$, $i = 1, 2$ for the two volatility factors at the original data points [21]. That is, the available data set now consists of $\{\tilde{y}_\tau\}_{\tau=1}^T$, $\{\tilde{V}_{1,\tau}\}_{\tau=1}^T$, and $\{\tilde{V}_{2,\tau}\}_{\tau=1}^T$, where $T$ is the length of the original observed data series. The latent volatility is now no longer latent but observable and available for analysis and forecasting.

## 3. Nordic/Baltic Electricity Market's Front Year and Front Quarter Contracts

The daily studies span more than 12 years, from the end of 2009 to the beginning of 2022 (April), resulting in approximately 3000 daily price movements for the front year and front quarter electricity price series (Supplementary Materials). Due to the non-stationarity of the price series, the analysis is based on stationary logarithmic price changes from the two series. Any evidence of effective SV-model market implementations implies random

price changes and a minimum of weak-form market efficiency. As a result, the markets can be used for both increased risk management and volatility (derivatives) measurements.

### 3.1. The Nasdaq OMX Front Year and Front Quarter Contracts

Summary statistics for the two time-series are presented in Table 1. Figure 1 reports the time series, distributions, and correlograms. Both the front year and the front quarter series have positive average price changes (positive drift). The standard deviation for the front year (1.521) is naturally lower than the front quarter (2.668), reporting lower risk. The maximum (9.6) and minimum (−9.7) numbers confirm lower risk for the front year relative to the front quarter (a maximum of 27.9 and a minimum of −14.5) contracts. The front year contracts have a negative skewness coefficient, suggesting that the return distributions are negatively skewed. In contrast, the front quarter contracts show a positive skewness, indicating a right-skewed distribution (more extreme positive price movements). Kurtosis coefficients for the first quarter series are much higher than zero (>>0), indicating a strongly peaked distribution with heavy tails. The front quarter contract series has a higher peak than the front year contract series, indicating that the quarter series contains even more observations near to the unconditional mean. The Cramer–von Mises normal test statistics [23] suggest non-normal return distributions. In contrast, the quantile normal test statistics suggest more normal distributed returns. Figure 1 plots (top and middle parts) visually support and display these findings. The serial correlation in the mean equation is high, and the Ljung-Box Q-statistic [24] for both series is substantial. Volatility clustering appears to be evident using the Ljung-Box test statistic for squared returns ($Q^2$) and ARCH test statistics. Non-stationary series are rejected by the ADF [25] and Phillips-Perron test statistics. The RESET [26] test statistic, which accounts for any deviation from the maintained model's assumptions, is noteworthy (instability). Finally, the BDS [27] test statistics show that all integrals (m) have extremely substantial data dependence. Figure 1 (bottom) shows correlograms for daily price and squared/absolute price changes up to lag 20. Correlograms for daily price changes reveal only modest dependency, but correlograms for squared and absolute returns show significant data dependence, mostly in the form of serial correlation. The price change (log returns) data series demonstrate that the amount of volatility appears to alter at random but has a time variable character, as is usual for financial markets. We also experimented with breaking trends in the movement equations, but our results suggested little evidence for trend breaks. Quandt-Andrews [28] and Bai and Perron [29] report insignificant statistics (Quandt-Andrews' [28] single breakpoint test static report for front year contract a max. Wald F-statistic (2 December 2016) 6.143415 {0.1510} and for front quarter contract a max. Wald F-statistic (17 March 2020) 5.187115 {0.2284}, and Bai and Perron's [29] multiple breakpoint tests report for front year contract 0 vs. 1 breaks 6.143415 with critical value: 8.58, and for front quarter contract 0 vs. 1 breaks 5.187115 with critical value: 8.58.) for single and multiple breakpoints for both contracts. The Value at Risk (VaR) is a well-known concept of measures of risk, and Table 1 includes the 2.5% and 1% VaR numbers for market participants.

**Table 1.** Nordic/Baltic Electricity Market Characteristics, 2010–2022.

| Panel A | Nasdaq OMX Front Year Return Series | | | | | | | | |
|---|---|---|---|---|---|---|---|---|---|
| Mean (all)/ M (-drop) | Median Std.dev. | Max./ Min. | Moment Kurt/Skew | Quantile Kurt/Skew | Quantile Normal | Cramer von-Mises | Serial dependence Q(12) | Q2(12) | VaR (1; 2.5%) |
| 0.03775 | 0.00000 | 21.5520 | 9.9166 | 0.28524 | 9.1610 | 1333.80 | 34.736 | 566.72 | −6.394% |
| 0.03716 | 2.10427 | −13.4348 | 0.35853 | 0.04656 | {0.0102} | {0.0000} | {0.0010} | {0.0000} | −4.452% |
| BDS-Z-statistic (*e* = 1) | | | | | Phillips & Perron | Augment DF-test | ARCH (12) | RESET (6;12) | CVaR (1; 2.5%) |
| m = 2 | m = 3 | m = 4 | m = 5 | m = 6 | −44.57339 | −44.6405 | 252.848 | 4.189406 | −8.389% |
| 10.7530 | 12.7224 | 15.3246 | 18.0319 | 0.11584 | {0.0000} | {0.0000} | {0.0000} | {0.0000} | −6.517% |
| {0.0000} | {0.0000} | {0.0000} | {0.0000} | {0.1246} | | | | | |

**Table 1.** *Cont.*

| Panel B | Nasdaq OMX Front Quarter Return Series | | | | | | | | |
|---|---|---|---|---|---|---|---|---|---|
| Mean (all)/ | Median | Max./ | Moment | Quantile | Quantile | Cramer- | Serial dependence | | VaR |
| M (-drop) | Std.dev. | Min. | Kurt/Skew | Kurt/Skew | Normal | von-Mises | Q(12) | Q2(12) | (1; 2.5%) |
| 0.01660 | 0.00000 | 27.8948 | 8.3326 | 0.20601 | 4.3358 | 12267.80 | 59.546 | 419.55 | −9.937% |
| 0.01176 | 3.32033 | −21.2991 | 0.50813 | −0.00653 | {0.1144} | {0.0000} | {0.0000} | {0.0000} | −7.038% |
| BDS-Z-statistic (*e* = 1) | | | | | P&Perron | Augment | ARCH | RESET | CVaR |
| m = 2 | m = 3 | m = 4 | m = 5 | m = 6 | I + Trend | DF-test | (12) | (12;6) | (1; 2.5%) |
| 10.9471 | 13.1693 | 15.1492 | 17.1172 | 0.30024 | −42.93681 | −42.9187 | 20.740 | 9.5316 | −12.516% |
| {0.0000} | {0.0000} | {0.0000} | {0.0000} | {0.0050} | {0.0000} | {0.0000} | {0.0000} | {0.0000} | −9.990% |

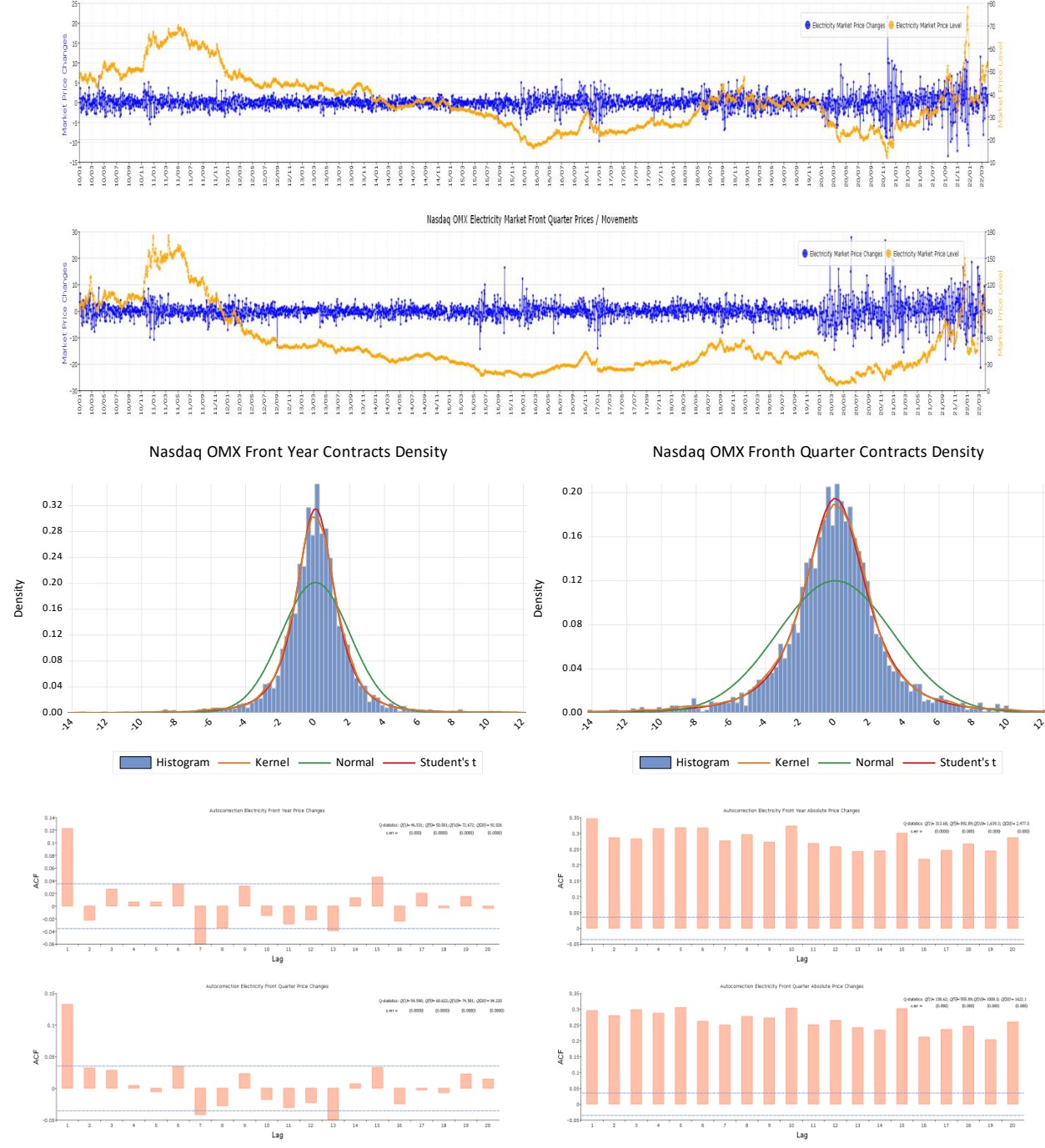

**Figure 1.** Nasdaq OMX Front Year and Front Quarter Contracts for the period 2010–2022.

### 3.2. Empirical Results

The conditional moments are calculated using a statistical model for density $(f(y|x))$, where $y$ represents price fluctuations and $x$ represents series delays (SNP). The stochastic volatility model (SV) from the equations above is estimated using the efficient method of moments (EMM [30]), that is, MCMC-GMM, employing the SNP generated conditional moments. Table 2 reports the conditional moments (panel A) and the BIC [31] optimal SV model coefficients (Panel B). Panel A reports the conditional moments that are used for the SV model simulation procedure. Note that we apply a spline transformation to squash the conditional values of the time series, not altering the asymptotic properties of the SNP estimators. The requirement for a largest eigenvalue for the variance less than one does no longer holds under this spline transformation (less than two under spline). Panel A reports the mode and standard errors for the conditional moments of the front year and front quarter contracts. The optimal BIC values are 1.17109 and 1.17361 for the front year and front quarter, respectively. Panel B reports columns for the mode, the mean and the standard errors for the front year and front quarter SV models. The SV model generates appropriate model test statistics, which are shown at the bottom of Panel B of Table 2. The objective function accuracy for the front year and front quarter contracts is $-2.3$ and $-2.7$, respectively, with related $\chi^2$ test statistics of 0.51 (3 df) and 0.44. (3 df). Together with diagnostics for the conditional moments from the fourteen statistical SNP estimation (all < 1.6 not reported; for score diagnostics, a value greater than 2 indicates diagnostic failure), the $\chi^2$ numbers report success. Figure 2 shows the MCMC log-posterior pathways. The model does not fail the test of over-identified limitations at the 10% level, the chains are choppy, and the densities are near to normal, all of which indicate that the SV model is adequate for the two electricity market contracts. As a result of the calculated SV model, the long-simulated realization of the state vector creates a functional form of the conditional distribution.

**Table 2.** Nasdaq OMX Moments and Stochastic Volatility Coefficients, 2022.

| **Panel A** | **The SNP Electricity Markets Conditional Moments** | | | | | |
|---|---|---|---|---|---|---|
| Coeff. | Front Year | theta | Standard errors | Front Quarter | theta | Standard errors |
| *Hermite Polynoms* | | | | | | |
| $h_1$ | $a_0[1]$ | 0.03455 | 0.0236 | $a_0[1]$ | $-0.00107$ | 0.0196 |
| $h_2$ | $a_0[2]$ | 0.03593 | 0.0319 | $a_0[2]$ | $-0.11727$ | 0.0166 |
| $h_3$ | $a_0[3]$ | $-0.01690$ | 0.0118 | $a_0[3]$ | $-0.01688$ | 0.0129 |
| $h_4$ | $a_0[4]$ | 0.07703 | 0.0113 | $a_0[4]$ | 0.11397 | 0.0111 |
| *Mean Equation (Correlation)* | | | | | | |
| $h_5$ | $b_0[1]$ | $-0.05713$ | 0.0281 | $B[1,1]$ | $-0.00975$ | 0.0243 |
| $h_6$ | $B[1,1]$ | 0.09000 | 0.0208 | $B[1,1]$ | 0.09046 | 0.0211 |
| *Variance Equation (Correlation)* | | | | | | |
| $h_7$ | $R_0[1]$ | 0.06775 | 0.0137 | $R_0[1]$ | 0.09497 | 0.0145 |
| $h_8$ | $P[1,1]$ | 0.31726 | 0.0320 | $P[1,1]$ | 0.37250 | 0.0291 |
| $h_9$ | $Q[1,1]$ | 0.95031 | 0.0064 | $Q[1,1]$ | 0.94747 | 0.0069 |
| $h_{10}$ | $V[1,1]$ | $-0.11353$ | 0.0851 | $V[1,1]$ | $-0.00096$ | 511,882.37 |
| Model | sn | 1.10494833 | | | 1.09814366 | |
| selection | aic | 1.10945468 | | | 1.10265001 | |
| criterias: | bic | 1.12252347 | | | 1.1157188 | |
| Largest eigenvalue mean: | | | 0.0900003 | | | 0.0904637 |
| Largest eigenvalue variance: | | | 1.003750 | | | 1.036460 |

**Table 2.** *Cont.*

| Panel B | Front Contracts Parameter Values for Scientific Models | | | | | |
|---|---|---|---|---|---|---|
| Coeff. $\theta$ | Front Year Mode | Mean | Standard errors | Front Quarter Mode | Mean | Standard errors |
| $a_0$ | 0.07813 | 0.07050 | 0.04805 | 0.04688 | 0.05533 | 0.04808 |
| $a_1$ | 0.07813 | 0.09337 | 0.02092 | 0.08984 | 0.08910 | 0.02027 |
| $b_0$ | 0.56250 | 0.54483 | 0.17370 | 0.76562 | 0.69850 | 0.12565 |
| $b_1$ | 0.97656 | 0.91499 | 0.04284 | 0.98047 | 0.95517 | 0.03767 |
| $c_1$ | 0.0 | 0.0 | 0.0 | 0.0 | 0.0 | 0.0 |
| $s_1$ | 0.08594 | 0.12427 | 0.02831 | 0.08203 | 0.09375 | 0.02493 |
| $s_2$ | 0.16406 | 0.07667 | 0.05527 | 0.20703 | 0.18367 | 0.05950 |
| $r_1$ | 0.06250 | −0.02430 | 0.13552 | −0.03125 | 0.06765 | 0.23628 |
| $r_2$ | −0.12500 | −0.08900 | 0.33878 | −0.07813 | −0.17646 | 0.17803 |
| Distributed (no. of freedom) | | | $\chi^2(3)$ | | | $\chi^2(3)$ |
| Posterior at the mode | | | −2.5271 | | | −2.7826 |
| Chisq. test statistic | | | {0.4704} | | | {0.4264} |

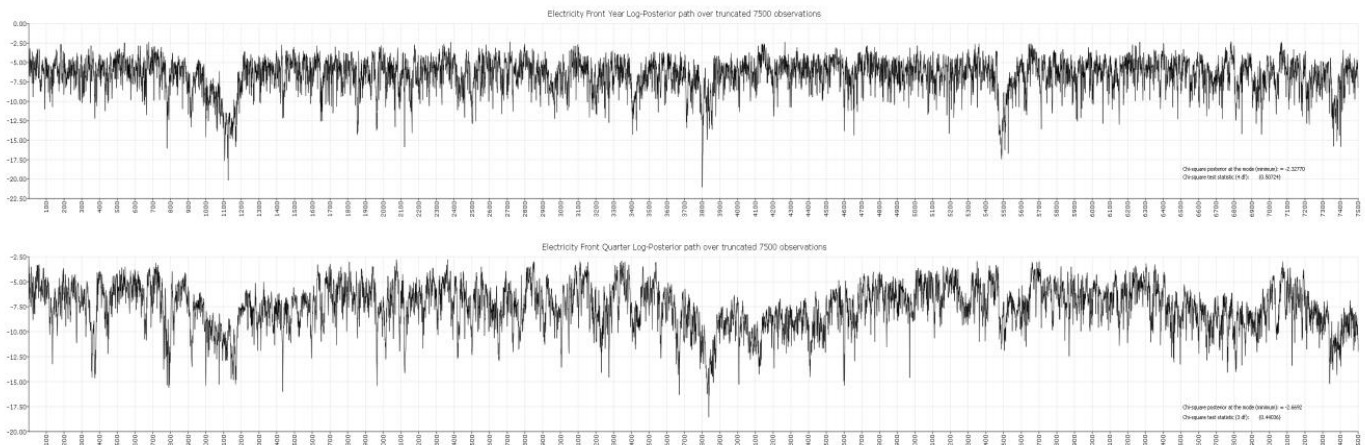

**Figure 2.** MCMC Posterior Chain from 250 $k$ Optimal SV Model (R = 75.000).

### 3.3. Nasdaq OMX Front Year and Front Quarter Stochastic Volatility

From the two recalibrated SNP models and their associated non-linear Kalman filter, the reprojected volatility for the observed dates is available $\left(\widetilde{V}_{it},\ i = 1, 2\right)$. The latent volatility is now no longer latent but observable and available for both analysis and forecasting. Figure 3 reports factor 1 ($V_1$), factor 2 ($V_2$) (left axis), and the $\sqrt{252}e^{(V_1+V_2)}$ (right axis) from the observed data points for the front year (top) and front quarter (bottom) contracts. Interestingly, $V_1$ is a slow-moving, persistent volatility factor, while $V_2$ is a fast-moving and strongly mean-reverting volatility factor. The volatility factors in Figure 3 seem to model two different flows of information to the electricity market and the market participants. Market transparency and the flow of information may therefore be classified according to factors. One slowly mean-reverting factor provides volatility persistence, and one rapidly mean-reverting factor provides for the tails [21]. Furthermore, from the volatility paths incorporating the projected yearly volatility $\sqrt{252}e^{(V_1+V_2)}$, the volatility seems more influenced by the $V_1$ persistent factor than the mean-reverting $V_2$ factor. The line for the $V_1$ factor is rarely exceeded by the $V_2$ factor (both measured on the left axis). Figure 3 also reports the ordinary least square $R^2$ number for $\hat{V}_{i,t}, i = 1, 2$ on $\hat{\sigma}_t^2$, $\hat{y}_t |\hat{y}_t|$ and generously long lags for the front year and the front quarter. For $V_1$ ($V_2$), the $R^2$ is 95.6% (4,8%) and 96.3% (5.9%) for the front year and front quarter, respectively. That is, the $V_1$ factor seems clearly easier to project than $V_2$ for both the front year and quarter contracts. The stochastic volatility $\left(\sqrt{252}e^{(V_1+V_2)}\right)$ for the front year and quarter is a combination



of their associated factors $V_1$ and $V_2$. The latent volatility is now immediately available and visible to all market participants. For electronic markets, the stochastic volatility series should be tradeable on a real-time basis.

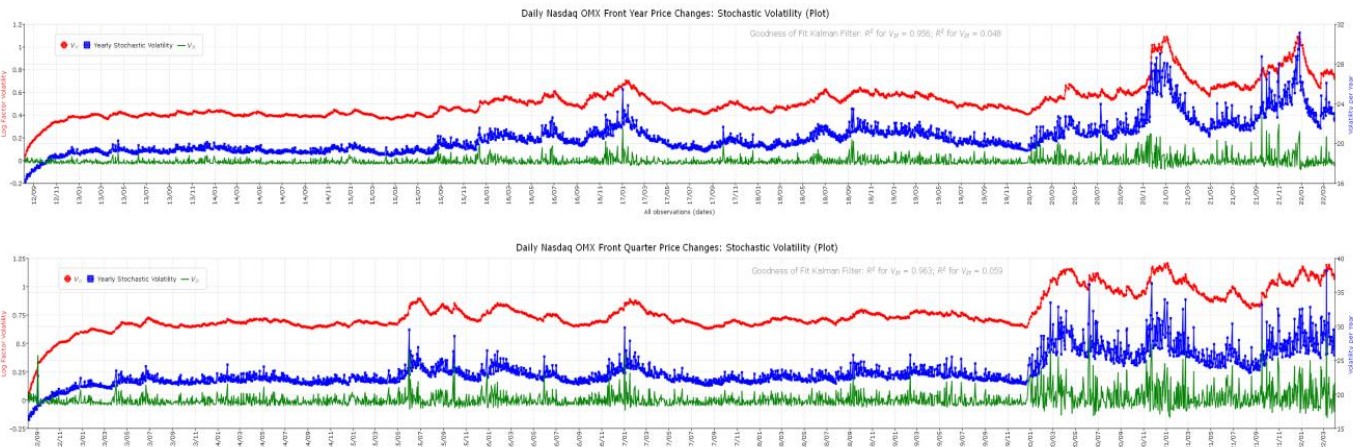

**Figure 3.** Stochastic Volatility from Observables using the Kalman Filter (daily).

When Figures 1 and 3 are compared, the two synchronous plots demonstrate that as returns become broader (narrower), volatility increases (decreases). Furthermore, turbulent (wide returns) days are more likely to be followed by other turbulent days, and calm (narrow returns) days are more likely to be followed by other tranquil (wide returns) days (clustering). Table 3 summarizes the volatility measures for the two electricity contracts. The volatility from the front quarter is larger than from the front year contracts. The $V_1$ ($V_2$) factor for the front year contracts reports a mean of 0.52 ($-0.002$) with a standard deviation of 0.14 (0.04). The higher volatility for the front quarter contracts is confirmed by a mean for $V_1$ ($V_2$) of 0.77 (0.004) and standard deviation of 0.16 (0.07). Moreover, due to the mean size differences for $V_1$ and $V_2$, the stochastic volatility for both contracts $\left( \sqrt{252} e^{(V_1 + V_2)} \right)$ seems dominated by the $V_1$ factor. The characteristics for $V_1$ will therefore most likely also be found for the stochastic volatility.

Table 3 shows the characteristics for $V_1$, $V_2$, and the stochastic volatility $\left( \sqrt{252} e^{(V_1 + V_2)} \right)$. The front year (front quarter) contracts' average volatility is 20.6 (23.5). The standard deviation is 1.7 for the front year and 2.4 for the front quarter contracts. Both contracts report non-normal densities for the Kalman filtered volatility. The Phillips–Perron and Augmented Dickey–Fuller tests cannot reject stationary volatility for the contracts. Moreover, a unit root with break test and with break selection minimizes the Dickey–Fuller t-statistic, rejecting the unit root for both front year and front quarter with $-13.54$ {<0.01} and $-21.38$ {<0.01}, respectively (probabilities in {}). Figure 4 reports the volatility densities (histogram) for the two contracts together with a Epanechnikov kernel and Student-t and Gamma distributions. Both contracts show a right-skewed distribution with a wider and higher density for the front quarter than the front year density.

**Table 3.** Nasdaq OMX Front Year and Quarter Volatility Characteristics 2020/2021.

| Panel A | Characteristics Nasdaq Front Year Contracts | | | | | | | |
|---|---|---|---|---|---|---|---|---|

**Volatility Factor $V_1$**

| Mean (all)/ Mode | Median Std.dev. | Maximum/ Minimum | Moment Kurt/Skew | Quantile Kurt/Skew | Quantile Normal | Cramer- von-Mises | Andersen Darling | Serial dep. Q(12) |
|---|---|---|---|---|---|---|---|---|
| 0.51887 | 0.48874 | 1.0954 | 2.43301 | 0.13542 | 10.8320 | 9.8053 | 63.85624 | 22713 |
| | 0.14465 | 0.0225 | 1.13207 | 0.14916 | {0.0044} | {0.0000} | {0.0000} | {0.0000} |
| BDS-Z-statistic ($e = 1$) | | | | | Phillips- Perron test | Augment DF-test | Breusch-Godfrey LM | |
| m = 2 | m = 3 | m = 4 | m = 5 | m = 6 | | | 10 lags | 20 lags |
| 124.121 | 146.201 | 175.798 | 218.816 | 280.884 | −3.41164 | −3.1497 | 2397.20 | 2397.31 |
| 0.00000 | {0.0000} | {0.0000} | {0.0000} | {0.0000} | {0.0107} | {0.0232} | {0.0000} | {0.0000} |

**Volatility Factor $V_2$**

| Mean (all)/ Mode | Median Std.dev. | Maximum/ Minimum | Moment Kurt/Skew | Quantile Kurt/Skew | Quantile Normal | Cramer- von-Mises | Andersen Darling | Serial dep. Q(12) |
|---|---|---|---|---|---|---|---|---|
| −0.00272 | −0.01341 | 0.3881 | 17.62417 | 0.08493 | 14.3653 | 33.1623 | 181.8012 | 374.23 |
| | 0.03744 | −0.0794 | 3.35739 | 0.18380 | {0.0008} | {0.0000} | {0.0000} | {0.0000} |
| BDS-Z-statistic ($e = 1$) | | | | | Phillips - Perron test | Augment DF-test | Breusch-Godfrey LM | |
| m = 2 | m = 3 | m = 4 | m = 5 | m = 6 | | | 10 lags | 20 lags |
| 12.395 | 13.329 | 14.954 | 16.204 | 17.577 | −51.840 | −9.6462 | 180.953 | 224.657 |
| 0.00000 | {0.0000} | {0.0000} | {0.0000} | {0.0000} | {0.0001} | {0.0000} | {0.0000} | {0.0000} |

**Reprojected Volatility ($\exp(V_1 + V_2)$)**

| Mean (all)/ Mode | Median Std.dev. | Maximum/ Minimum | Moment Kurt/Skew | Quantile Kurt/Skew | Quantile Normal | Cramer- von-Mises | Andersen Darling | Serial dep. Q(12) |
|---|---|---|---|---|---|---|---|---|
| 20.61478 | 20.20185 | 31.1449 | 4.29946 | 0.14271 | 11.0575 | 13.8382 | 84.6297 | 20112 |
| | 1.71739 | 16.1142 | 1.66877 | 0.14933 | {0.0040} | {0.0000} | {0.0000} | {0.0000} |
| BDS-Z-statistic ($e = 1$) | | | | | Phillips- Perron test | Augment DF-test | Breusch-Godfrey LM | |
| m = 2 | m = 3 | m = 4 | m = 5 | m = 6 | | | 10 lags | 20 lags |
| 86.772 | 99.537 | 115.601 | 137.650 | 168.371 | −8.73293 | −3.1274 | 2202.33 | 2205.25 |
| 0.00000 | {0.0000} | {0.0000} | {0.0000} | {0.0000} | {0.0000} | {0.0247} | {0.0000} | {0.0000} |

| Panel B | Characteristics Nasdaq Front Quarter Contracts | | | | | | | |
|---|---|---|---|---|---|---|---|---|

**Volatility Factor $V_1$**

| Mean (all)/ M (-drop) | Median Std.dev. | Maximum/ Minimum | Moment Kurt/Skew | Quantile Kurt/Skew | Quantile Normal | Cramer- von-Mises | Andersen Darling | Serial dep. Q(12) |
|---|---|---|---|---|---|---|---|---|
| 0.76788 | 0.71704 | 1.20739 | 1.56322 | 0.53077 | 122.3606 | 19.5508 | 104.339 | 26513 |
| | 0.16247 | 0.02150 | 0.33583 | 0.48238 | {0.0000} | {0.0000} | {0.0000} | {0.0000} |
| BDS-Z-statistic ($e = 1$) | | | | | Phillips- Perron test | Augment DF-test | Breusch-Godfrey LM | |
| m = 2 | m = 3 | m = 4 | m = 5 | m = 6 | | | 10 lags | 20 lags |
| 108.216 | 127.281 | 152.794 | 189.844 | 243.259 | −4.15052 | −4.2901 | 2390.69 | 2390.84 |
| {0.0000} | {0.0000} | {0.0000} | {0.0000} | {0.0000} | {0.0008} | {0.0005} | {0.0000} | {0.0000} |

**Volatility Factor $V_2$**

| Mean (all)/ M (-drop) | Median Std.dev. | Maximum/ Minimum | Moment Kurt/Skew | Quantile Kurt/Skew | Quantile Normal | Cramer- von-Mises | Andersen Darling | Serial dep. Q(12) |
|---|---|---|---|---|---|---|---|---|
| 0.00393 | −0.01455 | 0.56434 | 12.78224 | 0.20480 | 34.3745 | 26.4854 | 146.889 | 290.69 |
| | 0.06761 | −0.15333 | 2.83930 | 0.27326 | {0.0000} | {0.0000} | {0.0000} | {0.0000} |
| BDS-Z-statistic ($e = 1$) | | | | | Phillips - Perron test | Augment DF-test | Breusch-Godfrey LM | |
| m = 2 | m = 3 | m = 4 | m = 5 | m = 6 | | | 10 lags | 20 lags |
| 12.4713 | 16.0226 | 18.1589 | 20.1589 | 22.4214 | −55.20717 | −9.8673 | 154.1516 | 190.1991 |
| {0.0000} | {0.0000} | {0.0000} | {0.0000} | {0.0000} | {0.0001} | {0.0000} | {0.0000} | {0.0000} |

**Stochastic Yearly Volatility ($\sqrt{252}*\exp(V_1 + V_2)$)**

| Mean (all)/ M (-drop) | Median Std.dev. | Maximum/ Minimum | Moment Kurt/Skew | Quantile Kurt/Skew | Quantile Normal | Cramer- von-Mises | Andersen Darling | Serial dep. Q(12) |
|---|---|---|---|---|---|---|---|---|
| 23.46186 | 22.72147 | 38.20854 | 3.27852 | 0.36575 | 76.9821 | 22.3128 | 116.342 | 19360 |
| | 2.37134 | 16.27464 | 1.40411 | 0.39657 | {0.0000} | {0.0000} | {0.0000} | {0.0000} |
| BDS-Z-statistic ($e = 1$) | | | | | Phillips- Perron test | Augment DF-test | Breusch-Godfrey LM | |
| m = 2 | m = 3 | m = 4 | m = 5 | m = 6 | | | 10 lags | 20 lags |
| 70.6365 | 81.9115 | 95.5908 | 114.165 | 139.986 | −21.88660 | −2.9438 | 1955.24 | 1959.39 |
| {0.0000} | {0.0000} | {0.0000} | {0.0000} | {0.0000} | {0.0000} | {0.0406} | {0.0000} | {0.0000} |

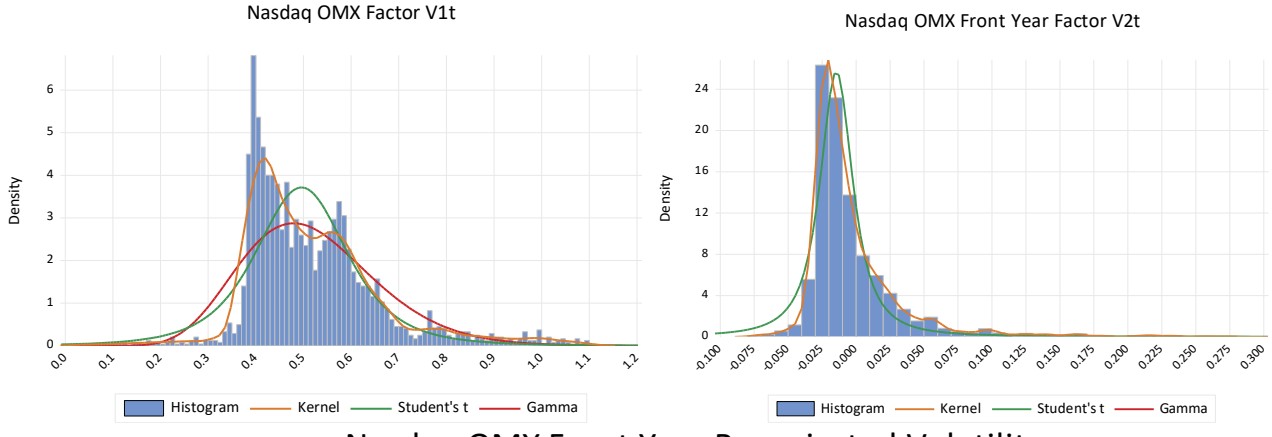

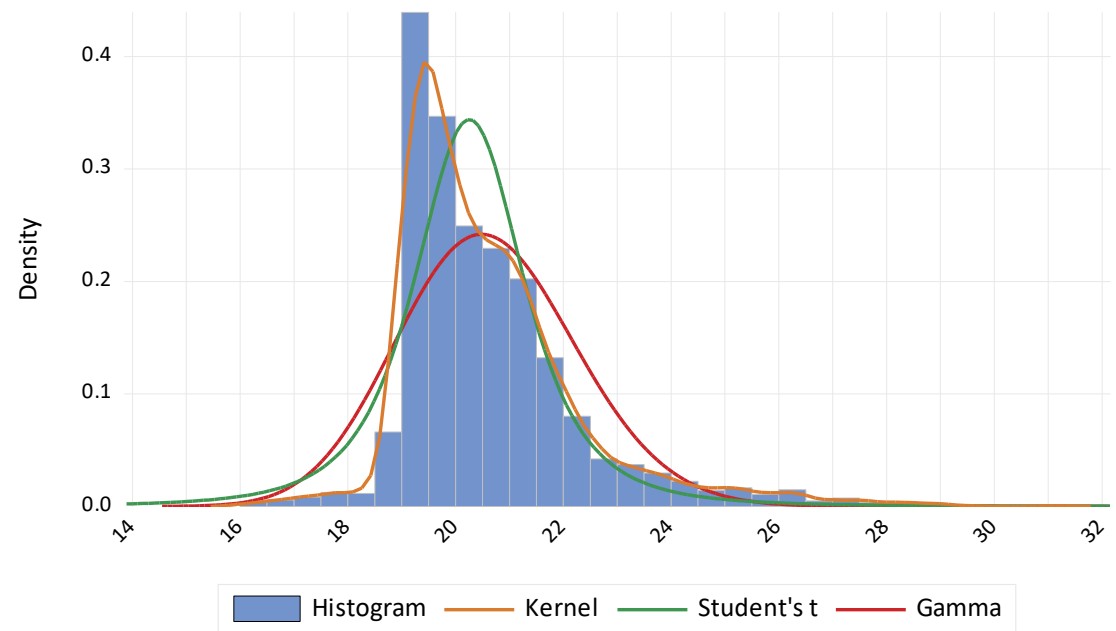

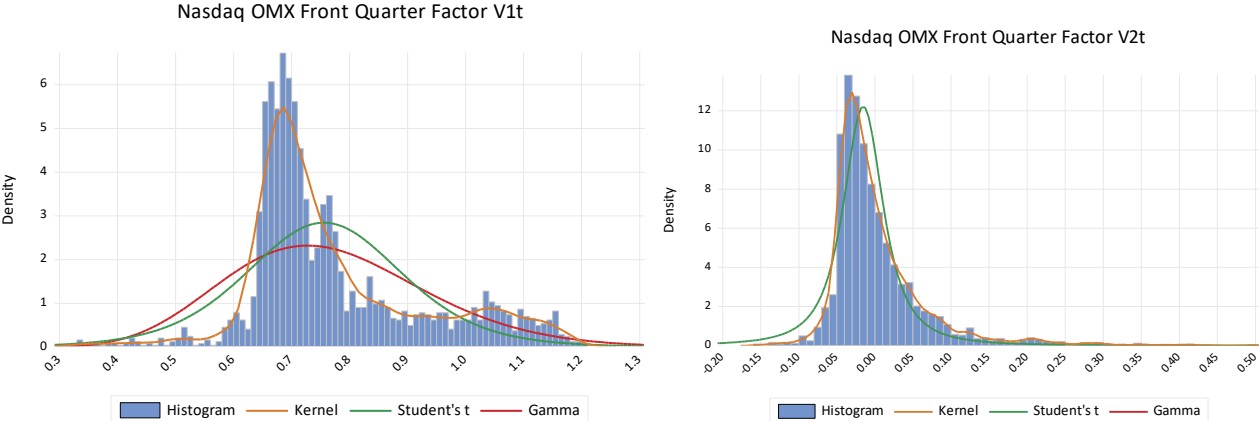

**Figure 4.** *Cont.*

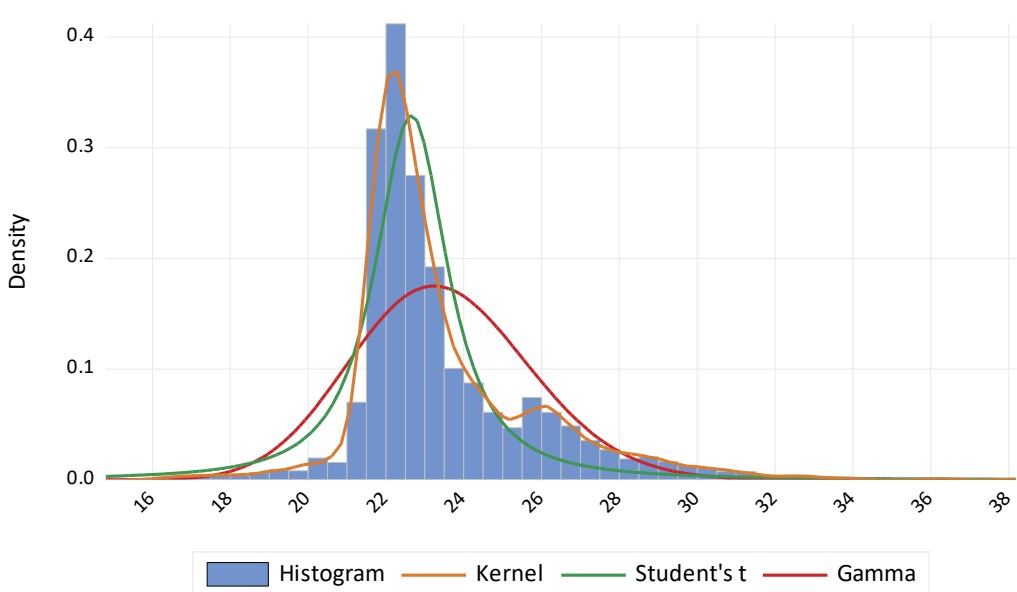

**Figure 4.** Stochastic Volatility Densities from Observables and the Kalman Filter.

For this application of the stochastic volatility model, an estimate of $y_t^* = \int\limits_{t}^{t+T} \exp(V_{1t} + V_{2t}) \cdot dt$, where $y_t^*$ is the contemporaneous latent variable, for the purpose of pricing options for the electricity contracts. Using Black and Scholes or the Binomial Formula to price options—for example, varying contract prices and time to maturity—may be beneficial for market participants. Figure 5 reports a Call–Put plot over moneyness (ln(S/K)) for the front year option contracts per 1 April 2022.

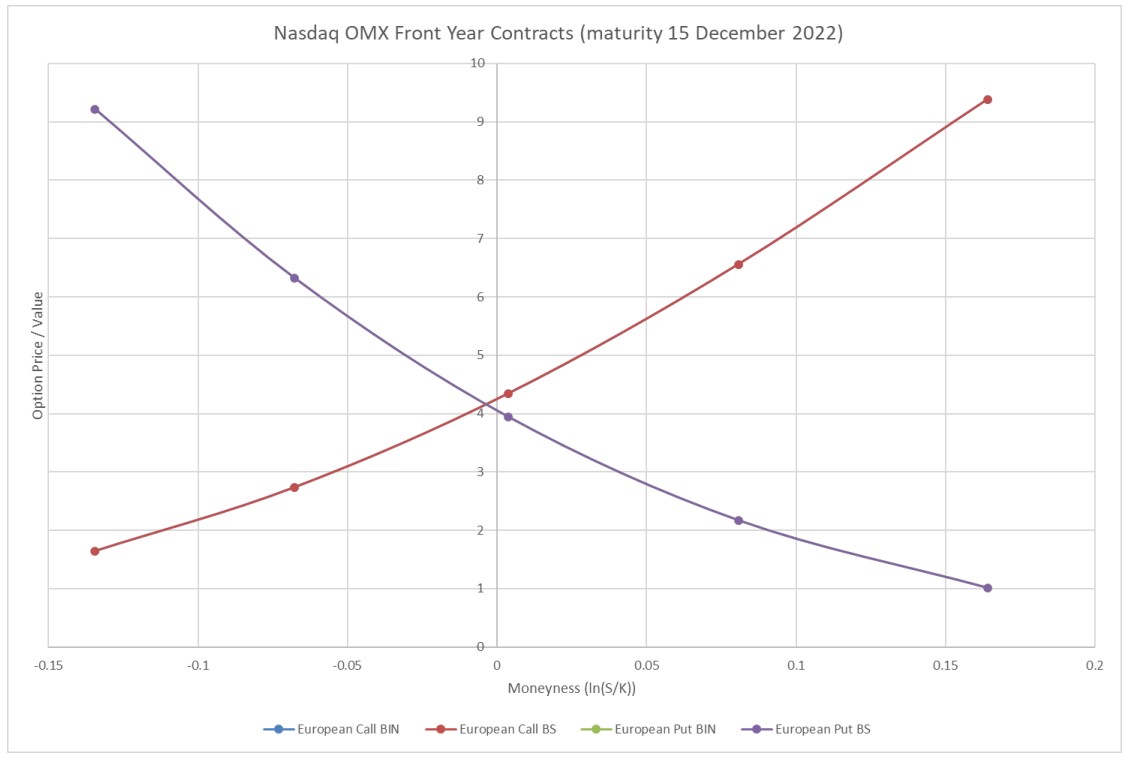

**Figure 5.** Option Prices over Moneyness for the Front Year Contracts.

### 3.4. Forecasting Nasdaq OMX Front Year and Front Quarter Volatility

For the front year serial correlation (*Q*) [32], the BDS Z-statistics [27] and Breusch–Godfrey [33] LM statistics in Table 3 suggest strong serial correlation (data dependence). Figure 6 reports the correlograms. The measures suggest substantial serial correlation (data dependence), that is, persistence (long memory). Forecasting volatility may therefore become plausible. Forecasting is challenging because the outcome of a stochastic process is influenced by random events that occur in the future. If there is a significant market movement before the risk horizon, the prediction must account for it. Static forecasts, on the other hand, are used to make a series of one-step-ahead forecasts of the two dependent variables, Nasdaq OMX front year and front quarter. In a static forecast, the methodology employs the actual value of the lagged endogenous variable for each observation in the forecast sample, requiring that data for both the exogenous and any lagged endogenous variables be observed for every observation in the forecast sample. Forecasting a realization of a stochastic process is difficult because the process will be influenced by random events that happen in the future. If there is a substantial market movement before the risk horizon, the prediction must account for it.

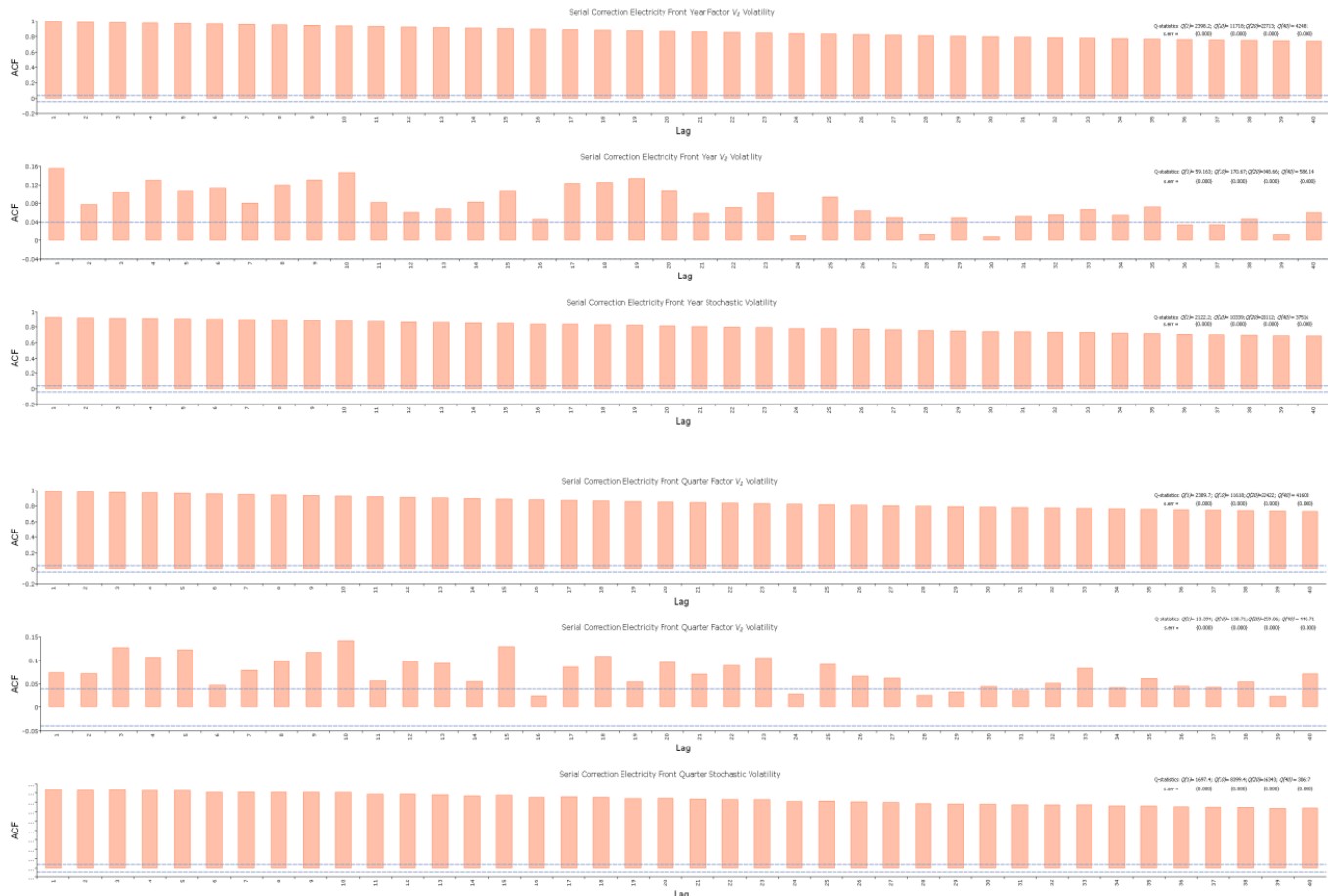

**Figure 6.** Stochastic Volatility Correlograms from Observables using the Kalman Filter.

For the market forecasting, the SV model is estimated for the period 2010–2021 (in-the-sample). The period 2021 to 2022 (4) is the out-of-sample period. Hence, the forecasting period runs from 1 January 2021 to 1 April 2022, with the estimating period for the SV model being from 2010 to 1 January 2021. Note that in practice, the SV model estimation can be estimated every week/month to incorporate the latest information in the model for forecasting purposes. (The SV model does not change significantly during the period 2021 to 2022 (4). For exposition and reporting, the SV model is therefore not updated during

the 2021 to 2022 (4) period. In practice, the SV model's weekly/monthly update will most certainly be performed for updated information). The RMSE and MAE are scale dependent on the dependent variable. The lower the inaccuracy, however, the higher the predicting ability. The MAPE and Theil measurements are not affected by scale. Theil's inequality coefficient is 0 for a perfect match. Using the Theil inequality coefficient (bias, variance, and covariance parts), for an "excellent" measure of fit, the bias and variance should be modest, with the majority of the bias focusing on the covariance percentage. Table 4 shows static forecast metrics for Nasdaq OMX contracts, and Figure 6 shows forecast plots with 95 percent confidence intervals for both factor components and the annual volatility. These forecasts confirm the benefits of volatility projections giving estimates of validity of using stochastic volatility models. For this application, pricing options are the main example.

**Table 4.** Nasdaq OMX Front Quarter and Front Year Contracts Static Forecast Evaluation Statistics 2020/2021.

| Estimated Daily Stochastic Volatility Forecast Fit Measures 01/21–04/22: | | | | | | | |
|---|---|---|---|---|---|---|---|
| Contracts: | Error Measures: | Factor 1 $V_1t$: | | Factor 2 $V_2t$: | | Stochastic Volatility ($e^{(V_1+V_2)}$): | |
| | Root Mean Square Error (RMSE) | 0.01680 | | 0.05883 | | 0.89793 | |
| | Mean absolute Error (MAE) | 0.01240 | | 0.03891 | | 0.621243 | |
| | Mean absolute percent error (MAPE) | 1.58480 | | 216.2324 | | 2.55794 | |
| | Teil inequality coefficient ($U_1$) | 0.01088 | | 0.74757 | | 0.01916 | |
| **Front Year** | Bias proportion | | 0.00022 | | 0.00196 | | 0.02857 |
| **Contracts:** | Variance Proportion | | 0.00330 | | 0.53616 | | 0.02857 |
| | Covariance Proportion | | 0.99648 | | 0.46189 | | 0.97119 |
| | Theil $U_2$ Coefficient | 0.96810 | | 1.67533 | | 0.85152 | |
| | Symmetric MAPE | 1.59258 | | 148.6957 | | 2.57765 | |
| | Root Mean Square Error (RMSE) | 0.01739 | | 0.10837 | | 1.77203 | |
| | Mean absolute Error (MAE) | 0.01306 | | 0.07922 | | 1.27864 | |
| | Mean absolute percent error (MAPE) | 1.27670 | | 153.421 | | 4.60947 | |
| | Teil inequality coefficient ($U_1$) | 0.00866 | | 0.74741 | | 0.03320 | |
| **Front Quarter** | Bias proportion | | 0.00050 | | 0.00456 | | 0.00051 |
| **Contracts:** | Variance Proportion | | 0.00041 | | 0.59765 | | 0.14362 |
| | Covariance Proportion | | 0.99909 | | 0.39779 | | 0.85587 |
| | Theil $U_2$ Coefficient | 0.98607 | | 1.14376 | | 0.76472 | |
| | Symmetric MAPE | 1.28205 | | 151.934 | | 4.66220 | |

## 4. Discussion

### 4.1. Stochastic Volatility Characteristics

The past years have seen an increasing popularity in trading volatility-based financial instruments such as VIX (VIX: The Chicago Board of Exchange Volatility Index (VIX), a real-time index that represents the market's expectations for the relative strength of near-term price changes of the S&P 500 Index in New York, NY, USA), perhaps due to its negative correlation with the equity instruments. Several researchers have studied the possibilities of reducing risk by including VIX and VIX-mimicking assets such as VIX futures and VIX options in their asset portfolio [34]. However, using VIX-mimicking portfolios does not provide a long-term hedge against rising volatility that most investors would desire [35]. Furthermore, Bordonado, ref. [36,37], touches upon the ineffectiveness of using VIX and VIX-based products as hedging instruments. This research, however, infers the volatility from logarithmic price movements modeled by a multiple-factor C/C++ unique model. The international research based on the unique SV model is therefore limited. The idea is to come up with alternative hedging instruments and strategies that will potentially be desirable for market participants.

The stochastic volatility characteristics for the front year and front quarter contracts from 2010 to 2022 are reported for $V_1$, $V_2$, and $e^{(V_1+V_2)}$ in Table 3. Table 3 reports non-normality, serial correlation (persistence), mean reversion (reject unit-root), and strong data

dependence (long memory). The volatility therefore mimics all known financial market characteristics, adding insight market hedging and derivative trading in general. From the step-ahead volatility projection, the following characteristics are readily available. The volatility is clearly not normal. The densities are clearly skewed to the right, showing log-normal characteristics. The yearly volatility is clearly highest for the front quarter contracts (shorter contracts with summer, autumn, winter, and spring differences) and therefore expected to show the lowest $R^2$ for the non-linear Kalman filter. The non-linear Kalman filter shows an $R^2$ for the front year $V_1$ ($V_2$) of 95.6% (4.8%) and the front quarter $V_1$ ($V_2$) of 96.3% (5.9%), giving a high confidence in the volatility methodology for both contracts.

To evaluate some probabilistic measures of the volatility series, the power law, an alternative to assuming normal distributions, is applied to $\left( \sqrt{252} e^{(V_1 + V_2)} \right)$. The power law asserts that, for many variables, it is approximately true that the value of the variable $v$ has the property that when $x$ is large, $Prob(v > x) = Kx^{-\alpha}$, where $K$ and $\alpha$ are constants. The relationship implies that $\ln[Prob(v > x)] = \ln K - \alpha \ln x$, and a test of whether it holds is plotting $\ln[Prob(v > x)]$ against $\ln(x)$. The $\ln(x)$ and $\ln[Prob(v > x)]$ values for the two front electricity contracts demonstrate that the logarithm of the chance of a change of more than $x$ standard deviations is essentially linearly dependent in $\ln(x)$ for $x \geq 3$. As a result, the power law holds for the re-projected volatility in both contracts. Regressions show the estimates of $K$ and $\alpha$ are as follows: for front year (front quarter) contracts, $K = e^{2.9776}$ and $\alpha = 5.9573$ ($K = e^{1.56364}$ and $\alpha = 5.19569$). A probability estimate of a volatility greater than 3 (6) standard deviations is therefore $19.6396 \, x \, 3^{-5.9573} = 0.02823(2.8235\%)$ $\left( 19.6396 \, x \, 6^{-5.9573} = 0.000454(0.0454\%) \right)$ and $4.7762 \, x \, 3^{-5.19569} = 0.015853(1.5853\%)$ $\left( 4.7762 \, x \, 6^{-5.19569} = 0.0004326(0.04326\%) \right)$ for the front year and the front quarter contracts, respectively. As an alternative, the extreme value theory can be used [38]. The $u$ is set to the 95 percentiles of the filtered volatility series of front year ($u = 24.062$) and front quarter ($u = 28.342$). The front year reports optimal $\beta = 1.7456$ and $\xi = 0.0$, with a log-likelihood function maximum value of $-189.99$. The front quarter series has optimum $\beta = 1.8271$ and $\xi = 0.0$, with a maximum value for the log-likelihood function of $-195.58$. The optimizer result $\xi = 0.0$ is likely a sign that the tail of the distribution is not heavier than the normal distribution. According to the extreme value theory, the front year contracts' re-projected volatility likelihood being larger than 25 (30) is 2.94 percent (0.1678 percent). The VaR with a confidence level of 99 (99.9) percent is 26.88 (30.90). As a result, the 99.9% VaR estimate is approximately 0.992 times lower than the maximum historical re-projected volatility. The 99 (99.9) percent expected shortfall (ES) estimate is 28.63 (32.65). In the same vein, for the front year contracts, the unconditional probability for a volatility greater than 24.062 ($u$) evaluated at the 99 (99.9) percent VaR level is equal to 0.573 (0.0573) percent (in general, a wider confidence interval than from a normal distribution). The likelihood that the stochastic volatility for the first quarter contracts will be larger than 25 (30) is 31.384 (2.034) percent. The VaR with a confidence level of 99 (99.9) percent is 31.296 (35.504). As a result, the 99.9 percent VaR estimate is approximately 0.929 times more than the greatest historical filtered volatility for front quarter contracts. The 99 (99.9) percent ES estimate is 33.124 (35.504). Finally, the unconditional chance of volatility larger than 28.342 ($u$) evaluated at the 99 (99.9) percent VaR level for front quarter futures is equal to 0.5473 (0.05473) percent. Var and ES are efforts to give a single number that encapsulates the volatility tails, so providing market participants with an indicator of the risk to which they are exposed. The daily volatility more than 25 percent has a likelihood of 2.943 percent for the front year contracts. According to the front quarter contracts, a daily volatility of more than 25% has a probability of 31.384 percent. As a result, both the power law and the extreme value theory, applying probabilities and VaR/ES values, describe tail features for the densities implying market risk. Inverting the unconditional probability of volatility and placing a 1% limit on the change in the unconditional probability will provide market players with a list of accessible investment possibilities.

In contrast to existing volatility markets calculated from derivative markets, this paper's volatility projection can immediately be updated from market movements (returns).

Hence, volatility can be a real-time market instrument that is tradable as an asset class in its own right. For instance, arbitrage traders and hedge funds may take positions on different volatilities of the same maturities, and speculators may simply make a bet on future volatility. Moreover, the volatility and contract prices from these electricity contracts are negatively correlated. The front year (front quarter) contracts report a negative correlation between returns and volatility of $-0.234$ ($-0.154$). Adding volatility to an electricity portfolio will therefore provide market participants with excellent diversification. Moreover, since volatility tends to increase markedly during market crashes, an electricity portfolio insures market participants.

*4.2. Stochastic Volatility Forecasts*

The static forecasts in Figure 7 with fit measures in Table 4 report a reasonably good fit for the projected stochastic volatility and its two factors ($V_1$ and $V_2$). Step-ahead volatility information is beneficial for trading in the derivative market as well standalone volatility instruments. Using the Theil inequality coefficient, which is a number between 0 and 1, for $V_1$, $V_2$, and the stochastic volatility ($e^{(V_1+V_2)}$) where portions sum to one, the results indicate step-ahead fit reliability. For both contracts, the $V_1$ factor and the projected volatility report a Theil inequality coefficient lower than 0.033. In contrast, the $V_2$ factor reports a Theil inequality coefficient greater than 0.747. The immediate implication is predictability for $V_1$ and the stochastic volatility and low predictability for $V_2$. A closer look at the portions of Theil's inequality gives the following insights. For $V_1$ and the stochastic volatility, the bias portion (how far the mean of the forecast is from the mean of the actual series) is close to zero (zero indicates a perfect fit), the variance portion (how far the variation of the forecast is from the variation of the actual series) is close to zero, while the covariance proportion (measuring the remaining unsystematic forecast error) is close to one. For the stochastic volatility ($e^{(V_1+V_2)}$) and the $V_1$ factor, most of the bias therefore concentrates on the covariance portion, indicating an overall good fit. In contrast, the $V_2$ factor does not concentrate on the covariance portion. From Table 4, the front year (front quarter) reports a covariance portion for the stochastic volatility of 0.971 (0.856). Furthermore, for the main contributor to the re-projected volatility for both contract series, factor $V_1$, the covariance portion of the Theil inequality coefficient is closer to one. For the front year (front quarter), the $V_1$ factor shows a covariance portion of 0.996 (0.999). In contrast, the front year (front quarter) for the $V_2$ factor shows a Theil's inequality covariance portion as low as 0.462 (0.398). Hence, the predictability seems to originate from the $V_1$ factor, influencing the predicted stochastic volatility (see also the scaling in Figure 3).

From Figure 6 (top), the front year volatility predictions for $V_1$ are well inside the 95% confidence intervals (high covariance portion (0.993)). For the predicted stochastic volatility, there are periods of 95% confidence interval breaks. The covariance portion is lower (0.971), indicating actual markets movements somewhat larger than predicted. The extra noise in volatility stem from the $V_2$ factor. For the front quarter contracts in Figure 6 (bottom), the front quarter volatility predictions for $V_1$ are well inside the 95% confidence intervals (high covariance portion (0.999)). For the predicted stochastic volatility, there are several periods of 95% confidence interval breaks. The covariance portion is clearly lower (0.856), indicating actual markets movements larger than predicted. The lower covariance portion for the predicted stochastic volatility stems from the $V_2$ factor (noisier than for the front year contracts). However, note that as the delivery period for the front quarter contract is $\frac{1}{4}$ of the front year contract, it is therefore natural that the front quarter contract is more sensitive to new information, mostly observed in factor $V_2$.

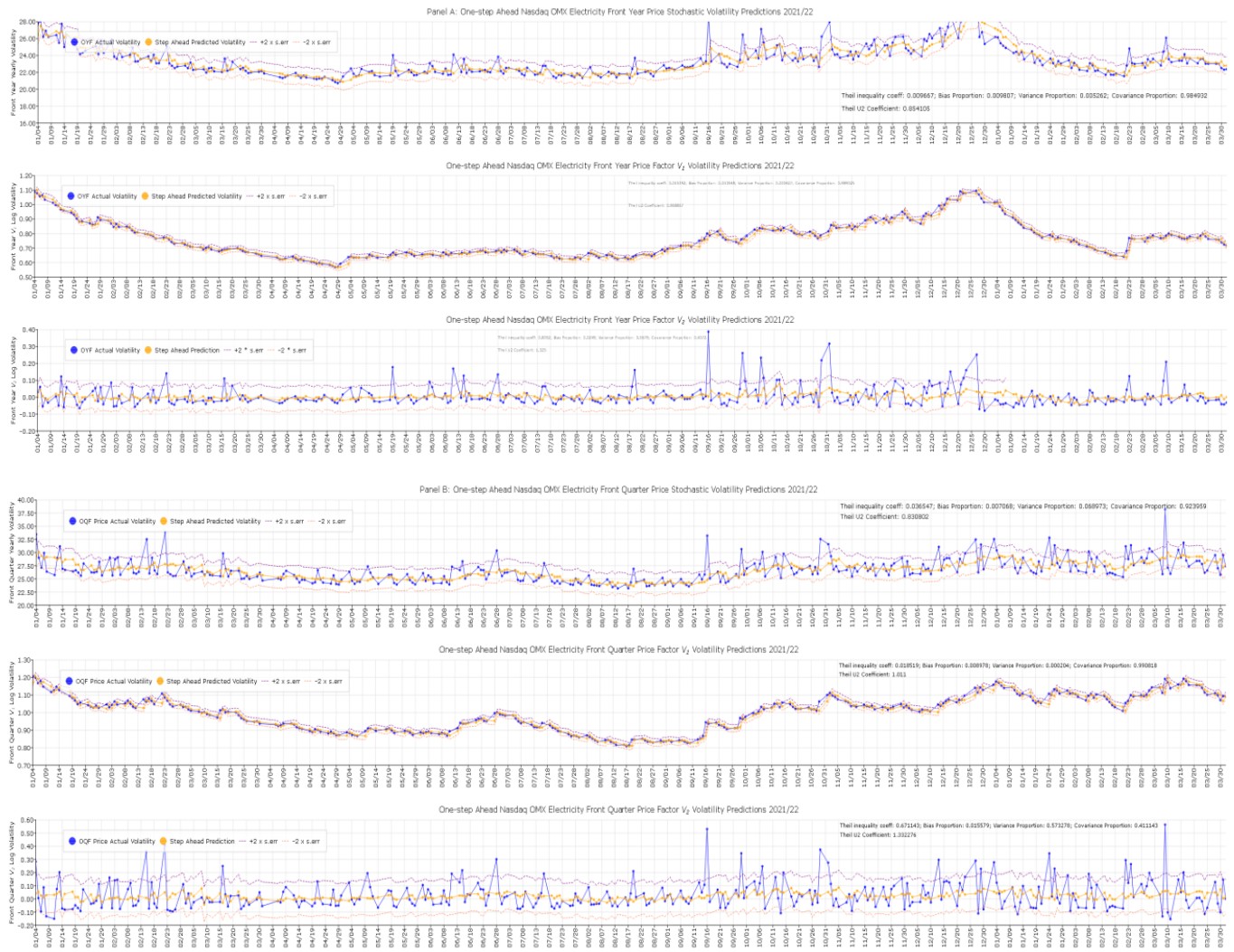

**Figure 7.** Nasdaq OMX Front Year (**top**) and Front Quarter (**bottom**) Contracts Static Forecasts for 2021/2022.

## 5. Conclusions

The major purpose of this work was to define a good volatility model as one that can predict and capture universally recognized stylized characteristics of financial market volatility. The volatility is inferred from asset returns. A great variety of stochastic volatility, deterministic volatility and jump models have been developed in recent years. Solibakke's [13,14] utilization of SV model applications is expanded upon in this work, applying logarithmic returns and the efficient method of moments (EMM) methodology [30]. Among the stylized realities for the volatility are heavy tails, persistence, mean reversion, asymmetry (negative return innovations indicate higher volatility), and long memory. The features imply that the volatility is highly dependent on data. The paper demonstrates that electricity market volatility possesses all these characteristics, and that the data dependence suggests that in this market, predictions can be used to improve risk management, portfolio timing and selection, market making, and derivative pricing for speculation and hedging.

This paper applies stochastic models relating volatility to risks that change through time in complicated ways. The departure from Black–Scholes–Merton option prices and occasional dramatic moves in markets is possible to explain (factors, correlation, data dependence). The paper shows that the stochastic volatility model separates into two distinct factors: a very persistent factor, $V_1$, showing low mean reversion, and a strongly mean-reverting factor, $V_2$. The persistent factor, $V_1$, provides for the main distribution,

and the rapidly mean-reverting factor, $V_2$, provides for the tails. The two-factor stochastic volatility model also reflects the shortcomings of the single-factor stochastic volatility model. A closer examination of the two Nasdaq OMX front year stochastic factors reveals that the persistent factor moves smoothly with an increase through COVID-19, and decreases in level toward the end of 2021/2022, whereas the strongly mean-reverting factor is choppy and reacts quickly to new information through the entire year of 2021/2022. For both contracts, the projected volatility seems to follow more the $V_1$ factor than $V_2$. The level of $V_1$ versus $V_2$ also confirms the $V_1$ influence on yearly volatility. Hence, knowing the different volatility factors may turn out to be an advantage for market participants.

Using an MCMC-GMM procedure of a multifactor stochastic volatility model with an associated Kalman filter procedure for reprojection makes the latent volatility visible. Volatility as an asset of its own will therefore extend investment strategies and make risk management both easier and more accessible. For instance, commodity volatility is strongly negatively correlated with prices, providing investors with excellent diversification and simultaneously insurance against market crashes. Moreover, the static forecasting exercise reports a high covariance portion from Theil's inequality coefficient. Trading volatility swaps and other derivatives may therefore become both accessible and less risky for all market participants.

Since the mid-1990s, many different stochastic volatility models have been proposed in the literature. By far the most popular model is the mean-reverting square root process from Heston [39]. This paper's stochastic volatility model research is an extended multifactor model inferred from logarithmic price movements. Chernov [12], Gallant and Tauchen [21], and Solibakke [13] have implemented and applied the model for equity markets. Inferring stochastic volatility from logarithmic price movements in financial markets is numerically inferior to volatility calculations from option markets (i.e., VIX). The model in Section 2 is unique [13] and implemented and programmed in C/C++. The methodology is computationally intensive and requires C/C++ programming experience. Prior international research is therefore limited. The research so far shows that volatility characteristics will be market-specific and dependent on the market's information flow, transparency, and participants.

Although pricing processes in energy markets are difficult to anticipate, the variance of prediction errors is obviously time dependent and appears to be estimable using observed previous fluctuations. The static forecasts for the electricity market contracts' projected volatilities reveal a Theil's inequality coefficient near to zero and a high covariance portion component. The major component for the projected volatility cycles, $V_1$, for the front year (quarter) contracts, has a Theil inequality coefficient covariance part of 99.3 (99.9%) percent. The tail component, $V_2$, for the contracts is considerably lower—about 5%. However, the $V_2$ component is much lower in size than $V_1$. Hence, the Theil's covariance portion for the front year (quarter) contracts is 97.1% (85.6%). The front quarter contracts have a more influential $V_2$ factor for the stochastic volatility. For the Nasdaq OMX electricity market players, a continuous SV model for projecting and forecasting, coupled with volatility trading strategies (i.e., derivatives and swaps), may enhance both assets and portfolios market strategies for all market participants.

The implemented and computerized SV model seems therefore to capture the full complexity of the generated volatility from price movements. This is natural in our research field, and we would normally assume that the price movements of an asset should contain the same information as option prices. This paper for a commodity market, together with international literature covering the equity market, has also shown that a multifactor stochastic volatility model reports success. There is still work to do covering implementation and computer efficiency for continuous updating of volatility/risk to financial markets.

**Supplementary Materials:** The following supporting information can be downloaded at: https://www.mdpi.com/article/10.3390/en15103839/s1. The article uses two freely available data sets ( 3000 observation). The first file "001-Electricity_Front_Year_prices_returns_2010–2022.txt" contains daily close financial contract prices (front year) in euros, freely downloadable from Nordpoolgroup.com (https://www.nordpoolgroup.com/Market-data1/#/nordic/table (accessed on 5 April 2022)). The second file "002-Electricity_Front_Quarter_prices_returns_2010–2022.txt" contains daily close financial contract prices (front quarter) in euros, freely downloadable from Nordpoolgroup.com (https://www.nordpoolgroup.com/Market-data1/#/nordic/table (accessed on 5 April 2022)). The Nordic/Baltic Electricity market consists of the following countries: Denmark, Estonia, Finland, Latvia, Lithuania, Norway, and Sweden. The Nord Pool Group has authorized the use of the data set. The consent is given under cite agreements, and the data should not be used without authorization. The https://www.nordpoolgroup.com/Market-data1/#/nordic/table (accessed on 5 April 2022) reference gives free and direct access to contract prices for the relevant period 2011–2022. The paper encloses the two daily data sets for the period 2010–2022: 1. 001-Electricity_Front_Year_Contract_ prices_returns_2010–2022.txt; 2. 002-Electricity_ Front_Quarter_Contract_prices_returns_2010–2022.txt.Elementary computer software for one-factor stochastic volatility models is available from: https://www.aronaldg.org/webfiles/ (accessed on 1 April 20221). Hardware computers (Linux OS) are from the Department of Industrial Economics and Technology Management, NTNU: http//www.ntnu.edu/iot (accessed on 1 April 2022).

**Funding:** This research received no external funding.

**Conflicts of Interest:** The author declares no conflict of interest.

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
