# Peer review of "Projecting and Forecasting the Latent Volatility for the Nasdaq OMX Nordic/Baltic Financial Electricity Market Applying Stochastic Volatility Market Characteristics"

_energies, doi:10.3390/en15103839_

Round 1

Reviewer 1 Report

 The manuscript is, in general, well written and organized and addresses an interesting subject.  In order to improve the quality of the article, I recommend several changes/adjustments, regarding the content and the format of the document, as follows:

  • Review the numbering of the papers in the Reference list, but also the way in which these papers are referred to in the article, so as to be in accordance with the rules regarding the paper format (the paper-works in the Reference list are not numbered either in alphabetical order or in the order of their appearance / reference in the text of the article).
  • Insert a Literature review section, for the analysis of recent research in the field and the identification of their main results, including the presentation of various Multifactor Stochastic Volatility Models. In section 2 (Materials and Methods) should be presented the model approached by the authors in the current research.
  • Justify the choice of the time-period analyzed (end of 2010 - beginning of 2021).
  • Mathematical relationships should be numbered so that they can be referred to more easily and explicitly.
  • Not all symbols used are explained in the models presented in Section 2.
  • Since section 2 has only one subsection, the latter does not need to be numbered.
  • In section 4 (Discussion) it would be recommended to discuss the results obtained in this study by comparing them with the results of other studies in the literature in the field.
  • In the Conclusions section, the added value of the research, the importance of the results and the way in which they can be used by the decision makers in formulating the policies on the Financial Electricity Market can be better highlighted.
  • The Conclusions section may indicate possible limitations of the study, as well as future research directions.

Reviewer 2 Report

 Discussion of results needs more comparison with prior studies. How are results similar or different from what has been done before in related papers? Moreover, I missed more explanations on the methodology part, in particular, regarding the steps they followed in their empirical results section. The authors should elaborate more on that.The Conclusion and result can be expanded. The results of the study should be specified.

Reviewer 3 Report

The paper deals with an important and interesting problem of modelling and forecasting volatility of electricity prices based on one year and one quarter forward prices from Nasdaq. The author uses a two factor stochastic volatility model to capture complex dynamics of analyzed series.

The model as well as estimation results are thoroughly (perhaps overly) described and discussed, although I would prefer fewer but better selected charts. Some of the included charts are barely readable. Moreover, I would expect a more precise positioning of the paper in the current literature on this area and a clearly defined contribution. The references included are mostly technical and/or general; almost nothing dealing with electricity market and its specifics. 

My main concern is however the forecasting part of the paper and its practical applicability. First of all, it is impossible to assess the quality of forecasts since no competing models are included. In order to address this, forecasting competitions are usually performed. Thus, supplementing results for some benchmark models well grounded in the literature  is a must. Moreover, as regards practical perspective, verifying forecasting quality  on realized/reprojected volatility (essentially nonobservable and nontradable as the author mentioned) is rather meaningless. Usually, an investor or regulator are interested in some specific risk measure (eg. VaR or ES) which is used to assess volatility/risk models in a purely out-of-sample fashion. This aspect is almost completely missing.

Finally, perhaps as a suggestion for future extensions, relating the results to volatility implied from options (which I believe are also available) would be very interesting and would made the analysis more comprehensive. 

Round 2

Reviewer 3 Report

The revised paper has been improved and some issues raised by the referees are addressed. However, I still do not see any effort to relate the results  to other studies on electricity market. Moreover, I remain unconvinced about the "forecasting" aspect of the paper since no out-of-sample verification (a.k.a backtesting) of the proposed volatility models is undertaken. Therefore, practical applicability claimed by the author is difficult to assess...

Author Response

attachment
